# Wealth disparities and economic flow: Assessment using an asset exchange model with the surplus stock of the wealthy

**Takeshi Kato**[1]*, **Yoshinori Hiroi**[2]

**1** Hitachi Kyoto University Laboratory, Open Innovation Institute, Kyoto University, Kyoto, Japan, **2** Kokoro Research Center, Kyoto University, Kyoto, Japan

\* kato.takeshi.3u@kyoto-u.ac.jp

## Abstract

How can we limit wealth disparities while stimulating economic flows in sustainable societies? To examine the link between these concepts, we propose an econophysics asset exchange model with the surplus stock of the wealthy. The wealthy are one of the two exchange agents and have more assets than the poor. Our simulation model converts the surplus contribution rate of the wealthy to a new variable parameter alongside the saving rate and introduces the total exchange (flow) and rank correlation coefficient (metabolism) as new evaluation indexes, adding to the Gini index (disparities), thereby assessing both wealth distribution and the relationships among the disparities, flow, and metabolism. We show that these result in a gamma-like wealth distribution, and our model reveals a trade-off between limiting disparities and vitalizing the market. To limit disparities and increase flow and metabolism, we also find the need to restrain savings and use the wealthy surplus stock. This relationship is explicitly expressed in the new equation introduced herein. The insights gained by uncovering the root of disparities may present a persuasive case for investments in social security measures or social businesses involving stock redistribution or sharing.

## Introduction

Income and regional disparities reflect an unequal distribution of wealth, and they have become a major social concern in the Organisation for Economic Co-operation and Development (OECD) countries and non-OECD countries such as those in Asia and Africa [1]. Goal 10 of the United Nations' Sustainable Development Goals targets a reduction in inequality within and among countries, while Goal 11 calls for making cities and human settlements inclusive, safe, resilient, and sustainable [2]. Income disparities cause poverty among people with low incomes and contribute to social unrest [3], whereas regional disparities lead to population maldistribution, as people flock to cities. The latter ultimately causes a decline in population and rapid aging in rural areas [4].

Econophysics scholars examine wealth distribution using multi-agent asset exchange models based on the analogy of the exchange of kinetic energy in a collision of two ideal gas particles [5, 6]. In an asset exchange model, an exponential distribution, a gamma distribution, a power law distribution, and a delta distribution emerge in accordance with parameters such as

**Funding:** This study was supported by the Humanities, Social/Behavioral Sciences, and Natural Sciences Interdisciplinary Research Project

of Kokoro Research Center, Kyoto University (http://kokoro.kyoto-u.ac.jp/category/researchproject/). The funder had no role in study design, data collection and analysis, decision to publish, or preparation of the manuscript.

**Competing interests:** The authors have declared that no competing interests exist.

the amount of exchange between the wealthy and the poor, wealth distribution, and saving rate. Thus far, scholars have proposed different models to examine the relationship between the parameters and distribution profile [7, 8]. We discuss Pareto and Zipf's laws in this context. Regarding income distribution, Pareto's law [9] states that the top 20% of income earners earn 80% of all income. Regarding city size distribution (population and scale), Zipf's law [10], which is also a power law, states that the size of the city that ranks $n$ is $1/n$ of the size of the top-ranking city.

The aim of this study is not only to examine wealth distribution but also to obtain guidelines for solving the problem of disparities. Hence, our objectives are to establish a new asset exchange model that covers the stock of the wealthy and to obtain a clear equation that can explain the relationship between wealth disparities and asset exchanges.

Therefore, unlike prior literature, we focus on the Gini index, total exchange, saving rate, and distribution shape and examine the relationships among them based on an asset exchange model. The Gini index (disparities) is expected to decline if the saving rate (stock) rises [6], but how do they affect the total exchange (flow)? If a model in which the wealthy and the poor engage in an equivalent exchange is compared with a model in which a non-equivalent exchange takes place with the wealthy contributing more, how do the two models identify the differences in disparities and flow? These questions form the basis for focusing on the relationships among stock, flow, and disparities to solve the disparities between the wealthy and the poor. Thus, this study theoretically creates a non-equivalent exchange model that focuses on the surplus stock of the wealthy, methodologically observes the level of market activity, and evaluates not only the wealth distribution and disparities but also the total exchange (flow) and rank correlation coefficient (metabolism). This methodology, which focuses on flow and metabolism, in addition to stock and disparities, is novel. Further, the relationship between them is represented by a unique equation. This study's new assessment method to limit disparities and vitalize flow will provide a fundamental rationale for the redistribution of stock from the viewpoint of econophysics.

The remainder of this paper is structured as follows: the next section presents a literature review, the "Methods" section proposes a new asset exchange model, and the "Results" section shows the results of the wealth distribution simulation and derives a new equation expressing the relationships among the total exchange, Gini index, saving rate, and surplus stock. The "Discussion" section covers the measures for disparities from the perspective of a stock redistribution policy and discusses the initiatives on abandoned farmland in rural districts and the revitalization of urban areas as part of this policy. The last section presents conclusions and future challenges.

## Literature review

This section provides a brief literature review based on Kato et al. [11]. In 1906, Italian economist Vilfredo Pareto was the first to observe the distribution of population and income in what became known as the Pareto principle [9]. This principle obeys a power law that dictates that income and wealth are concentrated among the wealthy (e.g., [12]). In 1953, Champernowne explained Pareto's law using a model that shows chronological changes in income distribution through a stochastic process [13]. In 2009, Yakovenko and Rosser, based on a review of many studies, suggested that the distribution of income and wealth matches a logarithmic normal distribution or a gamma distribution and that the distribution tails obey a power law [7].

Then, in 1986, John Angle, a sociologist, demonstrated that a gamma distribution emerges from a stochastic process model wherein economic agents randomly exchange their surplus assets other than savings [14]. From the standpoint of econophysics, Hayes [15] and Chakraborti [16] used a kinetic energy exchange model with a random collision of ideal gas particles

to demonstrate that a delta distribution emerges wherein wealth is concentrated in a single economic agent. This model differs from that of Angle in that it determines the amount of exchange in accordance with the poor.

The models by Angle, Hayes, and Chakraborti were further enhanced by later scholars, and several new models were proposed, such as an exponential distribution obtained from the model in which the exchange amount is randomly divided among economic agents [8]. In Chatterjee and Chakrabarti's model [6], a gamma distribution is obtained in which all economic agents set aside a certain percentage in saving. At the same time, a power law distribution is obtained from Chatterjee et al.'s model [5] wherein the saving rate of economic agents follows a uniform distribution.

Several other models have been proposed to limit wealth disparities. In Guala's model [17], a fixed tax rate is imposed on economic agents and the tax revenue is distributed to the economic agents. The exponential distribution changes to a gamma distribution and then becomes an exponential distribution again as the tax rate rises. In a different model [18], economic agents take out insurance to avoid risks, and the winner transfers the insurance payout to the loser. The exponential distribution changes to a gamma distribution and then to a delta distribution. As for regional disparities, Kato et al.'s model [11] includes a regional sphere (spatial exchange range) and a regional support bias (i.e., a distribution norm wherein assets are divided by assigning an advantageous probability previously to the poorer region rather than the wealthier region) in addition to the regional economic circulation rate (the saving rate). In this model, a distribution with large disparities undergoes changes and approaches a normal distribution.

Thus far, several asset exchange models that explain wealth distribution have been overviewed. The objective of this study is not to refine the traditional asset exchange models but to gain clear insights into the relationships among the saving rate (stock), total exchange (flow), and Gini index. In the models reviewed in this section, all the surplus assets, excluding savings, were regarded as the exchange amount between the wealthy and the poor, or the exchange amount between the two sides was determined in accordance with the surplus assets of the poor (twice the surplus assets of the poor). We propose a new model that allows greater flexibility in determining the exchange amount with respect to the saving rate to clarify the relationships among stock, flow, and disparities. In other words, in contrast to traditional models in which the surplus asset and the exchange amount are intertwined, we introduce, for the first time, a model that separates the two.

## Methods

This section refers to several conventional models to establish a unique non-equivalent exchange model that can handle the savings and surplus stock to evaluate stock, flow, metabolism, and disparities in a lucid manner.

In a general asset exchange model [5–8], two agents, $i$ and $j(= 1,2,\cdots,N)$, are randomly chosen from $N$ agents. The asset of agent $i$ at time $t$ is expressed as $m_i(t)$, while the asset of agent $j$ at time $t$ is $m_j(t)$. These two agents, $i$ and $j$, exchange some assets at a random division probability $\varepsilon$. The probability $\varepsilon$ is usually a uniform random number defined within the range of $0 \leq \varepsilon \leq 1$. The assets $m_i(t+1)$ and $m_j(t+1)$ at time $t+1$ are defined by a set of Eq (1). In this model, there is no distinction between the two agents.

$$m_i(t + 1) = \varepsilon \cdot (m_i(t) + m_j(t))$$
$$m_j(t + 1) = (1 - \varepsilon) \cdot (m_i(t) + m_j(t)) \tag{1}$$

where $\varepsilon$ is a uniform random number ($0 \leq \varepsilon \leq 1$).

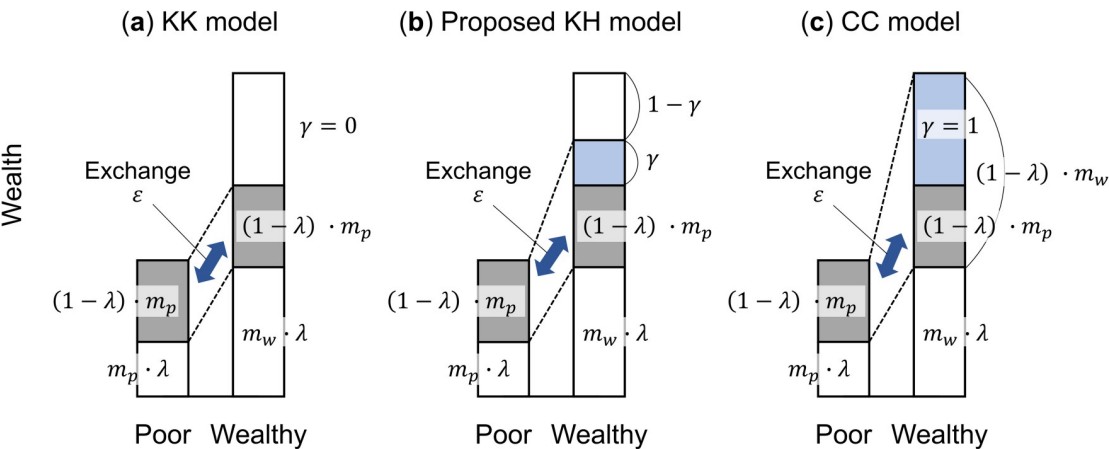

**Fig 1. Asset exchange models.** (a) Kato and Kudo (KK) model (the surplus contribution rate of the wealthy $\gamma = 0$), (b) proposed KH model (a combination of the Chatterjee and Chakrabarti [CC] and KK models ($0 \leq \gamma \leq 1$), and (c) CC model ($\gamma = 1$).

The model proposed by Chatterjee and Chakrabarti [6], which we call the "CC model" for convenience, uses savings. This model establishes a common saving rate $\lambda$, for $N$ agents. Two agents, $i$ and $j$, save a portion of their assets at the saving rate $\lambda$ at time $t$. The wealthy and poor were not distinguished in the CC model. They then exchange their surplus assets other than savings, $(1-\lambda) \cdot (m_i(t) + m_j(t))$, at the random division probability $\varepsilon$, shown in Fig 1C. The assets $m_i(t+1)$ and $m_j(t+1)$ at time $t+1$ are expressed in Eq (2). The exchange process of the CC model, if repeated, results in a gamma distribution, which has smaller disparities than the power distribution. In the CC model, as in Eq (1), there is no distinction between the two agents.

$$m_i(t+1) = \lambda \cdot m_i(t) + \varepsilon \cdot (1-\lambda) \cdot (m_i(t) + m_j(t))$$
$$m_j(t+1) = \lambda \cdot m_j(t) + (1-\varepsilon) \cdot (1-\lambda) \cdot (m_i(t) + m_j(t)) \qquad (2)$$

where $\varepsilon$ is a uniform random number ($0 \leq \varepsilon \leq 1$).

Meanwhile, the model proposed by Kato et al. [11], which we call the "KK model," distinguishes the wealthy and the poor. The exchange amount is determined in accordance with $(1-\lambda) \cdot \mathrm{Min}(m_i(t), m_j(t))$, the surplus assets of the poorer of the two agents, $i$ and $j$, after savings are excluded in Fig 1A. In Fig 1, the poor asset $\mathrm{Min}(m_i(t), m_j(t))$ is shown as $m_p$, and the wealthy asset $\mathrm{Max}(m_i(t), m_j(t))$ is shown as $m_w$. The assets $m_i(t+1)$ and $m_j(t+1)$ at time $t+1$ are expressed in Eq (3). The exchange process of the KK model, if repeated, gradually approaches a delta distribution, a situation wherein all the assets are concentrated in a single agent.

$$m_i(t+1) = m_i(t) - (1-\lambda) \cdot \mathrm{Min}(m_i(t), m_j(t)) + 2 \cdot \varepsilon \cdot (1-\lambda) \cdot \mathrm{Min}(m_i(t), m_j(t))$$
$$m_j(t+1) = m_j(t) - (1-\lambda) \cdot \mathrm{Min}(m_i(t), m_j(t)) + 2 \cdot (1-\varepsilon) \cdot (1-\lambda) \cdot \mathrm{Min}(m_i(t), m_j(t)) \quad (3)$$

where $\varepsilon$ is a uniform random number ($0 \leq \varepsilon \leq 1$).

We now compare the CC and KK models with respect to the surplus assets of the wealthy. In the CC model, the wealthy contribute all their assets to be exchanged, but in the KK model, they only contribute the same amount contributed by the poor. This may be why the CC model results in a gamma distribution with relatively small disparities and the KK model leads to a delta distribution with extremely large disparities [11].

However, in asset exchange, that is, economic transactions, it is not realistic to expect that the wealthy contribute all their assets as in the CC model [19, 20]. Meanwhile, if the wealthy engage in an equivalent exchange in accordance with the surplus assets of the poor, as in the KK model, extremely large disparities will emerge [21].

Thus, to limit disparities within a realistic range, it will be effective to combine the CC and KK models by incorporating the surplus asset redistribution of the wealthy into an asset exchange model. This combined approach is our original methodology, and this new model is called the "KH model." In the KH model, as shown in Fig 1B, the poor contribute their surplus assets, $(1-\lambda)\cdot m_p$ (the gray area). The wealthy contribute the amount equivalent to the surplus assets of the poor (the gray area) plus the amount obtained by subtracting the surplus assets of the poor from the surplus assets of the wealthy, multiplied by $\gamma$ (the blue area), $(1-\lambda)\cdot(m_p+\gamma\cdot(m_w-m_p))$. The parameter $\gamma$ was introduced for the first time in the KH model to specifically express stock redistribution, as proposed by Hiroi [22, 23], which is a policy proposal not supported by any mathematical model. Hiroi [22, 23] argued that stock is redistributed because the Gini index of housing and land assets (stock) is larger than the Gini index of income (flow) in Japan's statistical surveys.

The amount contributed by the wealthy and that by the poor will be exchanged at the random division probability $\varepsilon$ (the dark-blue arrow). Thus, the assets $m_i(t+1)$ and $m_j(t+1)$ at time $t+1$ are expressed in Eq (4). In the KK model, the surplus contribution rate of the wealthy is $\gamma = 0$, as shown in Fig 1A, while in the CC model, it is $\gamma = 1$, as shown in Fig 1C. Thus, the KK and CC models can be bridged by changing the surplus contribution rate $\gamma$ within the 0–1 range in the KH model.

$$m_p = \mathrm{Min}(m_i(t),\ m_j(t))$$
$$m_w = \mathrm{Max}(m_i(t),\ m_j(t))$$
if $m_i(t+1) \leq m_j(t+1)$.
$$m_i(t+1) = m_i(t) - (1-\lambda)\cdot m_p + \varepsilon\cdot(1-\lambda)\cdot(2\cdot m_p + \gamma\cdot(m_w - m_p));$$
$$m_j(t+1) = m_j(t) - (1-\lambda)\cdot(m_p + \gamma\cdot(m_w - m_p)) + (1-\varepsilon)\cdot(1-\lambda)\cdot(2\cdot m_p + \gamma\cdot(m_w - m_p)) \quad (4)$$
if $m_i(t+1) > m_j(t+1)$.
$$m_i(t+1) = m_i(t) - (1-\lambda)\cdot(m_p + \gamma\cdot(m_w - m_p)) + \varepsilon\cdot(1-\lambda)\cdot(2\cdot m_p + \gamma\cdot(m_w - m_p));$$
$$m_j(t+1) = m_j(t) - (1-\lambda)\cdot m_p + (1-\varepsilon)\cdot(1-\lambda)\cdot(2\cdot m_p + \gamma\cdot(m_w - m_p)),$$
where $\varepsilon$ is a uniform random number $(0 \leq \varepsilon \leq 1)$.

The Gini index $g$, which represents disparities, can be obtained by drawing a line of perfect equality and a Lorenz curve [24]. Specifically, the assets of $N$ agents at time $t$, expressed as $m_i(t)$ $(i = 1,2,\cdots,N)$, are arranged in order of amount and calculated by Eq (5). Sort$(m_i(t))$ means to sort $m_i(t)$ in non-decreasing order. For example, in a perfectly equal situation, as in the case of the initial distribution of assets, the Gini index $g$ becomes 0. In a completely unequal situation, where all wealth is concentrated in a single agent, as in the case of the delta distribution, the Gini index $g$ becomes 1.

$$g = \frac{2 \cdot \sum_{i=1}^{N} i \cdot r_i(t)}{N \cdot \sum_{i=1}^{N} r_i(t)} - \frac{N+1}{N}, r_i(t) = \mathrm{Sort}(m_i(t)). \quad (5)$$

Here, two new indexes are introduced. One is the total exchange $f$ obtained by aggregating the exchange amount of the wealthy and that of the poor at time $t$ in Eq (4), that is, $(1-\lambda)\cdot(2\cdot m_{p_t} + \gamma\cdot(m_{w_t} - m_{p_t}))$, from time $t = 1$ to time $t = t_{max}$. We introduce $f$, an

original parameter, for the first time in this study. This is expressed in Eq (6). The denominator in this equation is for standardization purposes and the numerator constitutes the total exchange amount produced when an asset exchange takes place between two agents at each unit of time from time $t = 1$ to time $t = t_{max}$. The larger the total exchange $f$, the more active the asset exchange. In such a situation, the market is highly activated, and the flow is large.

$$f = \frac{\sum_{t=1}^{t_{max}} (1 - \lambda) \cdot (2 \cdot m_{p_t} + \gamma \cdot (m_{w_t} - m_{p_t}))}{2 \cdot t_{max}}. \tag{6}$$

The other index is the Kendall rank correlation coefficient $\tau$ [25], which is commonly used to measure the ordinal association between two measured quantities. We also introduce $\tau$, another parameter, for the first time in this study, to evaluate economic metabolism. It is obtained from the number $K$, at which the magnitude relationship of the pair $(m_i(t_1), m_j(t_1))$ at time $t_1$ agrees with that of the pair $(m_i(t_2), m_j(t_2))$ at time $t_2$, and the number $L$, at which they do not agree. This is expressed in Eq (7). The denominator for Eq (7) is a binomial coefficient for selecting two agents from $N$ agents. The rank correlation coefficient $\tau$ is 1 if the ranking agreement is perfect, and 0 otherwise. The larger the value of $\tau$, the smaller the metabolism, even if the disparities are fixed.

$$\tau = \frac{K - L}{\frac{N \cdot (N-1)}{2}}. \tag{7}$$

In this way, a condition exists to allow us to examine the relationships among the savings $\lambda$ (stock), total exchange $f$ (flow), Gini index $g$ (disparities), and rank correlation coefficient $\tau$ (metabolism) by changing the saving rate $\lambda$ and the surplus contribution rate $\gamma$ of the wealthy.

## Results

### Wealth distribution

First, we use the KH model represented by Eq (4) to examine how wealth is distributed in accordance with the surplus contribution rate $\gamma$ of the wealthy. Fig 2 shows the simulation results of wealth distribution. The saving rate for all agents is $\lambda = 0.25$, which is set here as a parameter since the average saving rate per GDP in the world is approximately 0.25 [26]. In the case of $\gamma = 0$ (the KK model) in Fig 2A, the power law distribution undergoes changes and approaches a delta distribution with larger disparities as time $t$ passes. Changes from $\gamma = 0.1$ in Fig 2B, to $\gamma = 0.5$ in Fig 2C, to $\gamma = 1$ in Fig 2D (the CC model), indicate that the distribution gradually narrows and the disparities diminish. Considering these distributions as gamma distributions, the shape parameter $k$ or the scale parameter $\theta$ of the gamma distribution appears to increase [27]. That is, we find that the surplus contribution rate $\gamma$ is related to the scale $k$ or shape $\theta$ of the gamma distribution.

### Gini index, total exchange, and rank correlation coefficient

Next, we observe the changes in the Gini index in response to the passage of time (the number of exchanges) using Eq (5). Fig 3 shows the simulation results of the Gini index $g$. Two different saving rates, $\lambda = 0.4, 0.8$, which are experimentally set here as parameters for broad observations, and four different surplus contribution rates of the wealthy, $\gamma = 0, 0.1, 0.5, 1$ (for a total of eight cases), are simulated. In Fig 3, generally, the Gini index $g$ converges at a certain value as time $t$ passes. In the case of $\gamma = 0$ (the KK model), it has not completely converged at $t = 10^6$; as time $t$ passes, the Gini index $g$ converges at 1, a delta distribution. Meanwhile, the larger the saving rate $\lambda$ and the surplus contribution rate $\gamma$ of the wealthy, the smaller the Gini index $g$ and the disparities. When

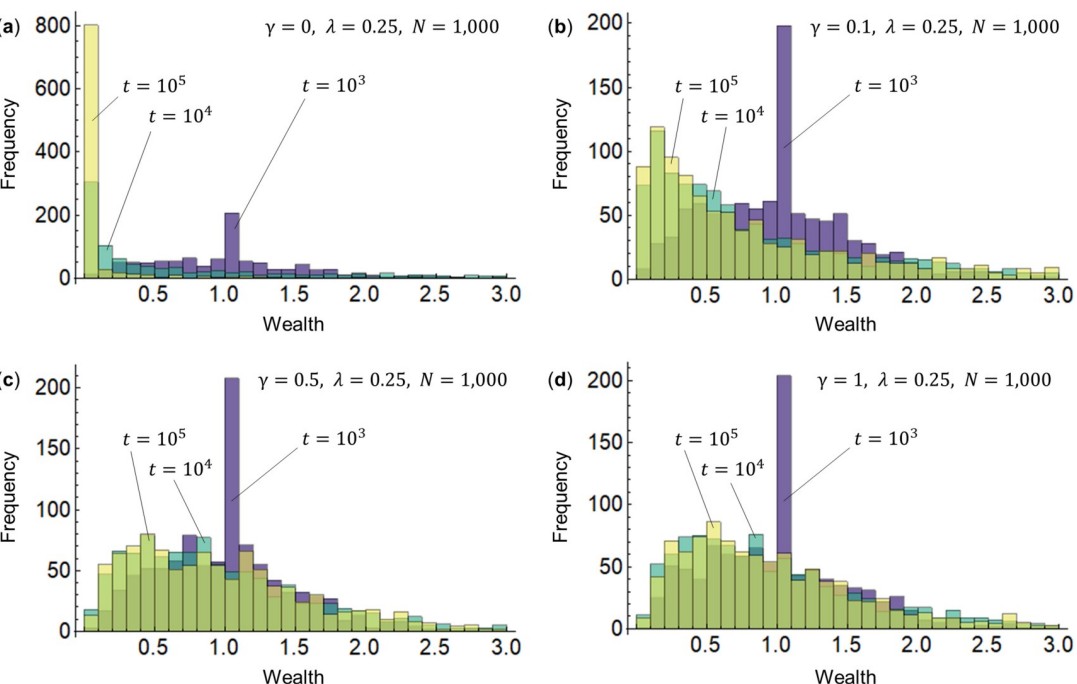

**Fig 2. Wealth distribution for the surplus contribution rate $\gamma$ of the wealthy.** The horizontal axis represents wealth (assets) while the vertical axis represents frequency. The number of agents is set at $N = 1,000$. The initial value for assets at time $t = 0$ for all agents is $m_i(0) = 1 (i = 1,2,\cdots,N)$. The saving rate for all agents is $\lambda = 0.4$. The four different surplus contribution rates of the wealthy are $\gamma = 0, 0.1, 0.5, 1$. For each case, we observe the changes in the distribution of wealth when the time (the number of changes) is $t = 10^3, 10^4$, and $10^5$.

the same saving rate $\lambda$ is used for comparison, the disparities decrease in the following order: the KK model ($\gamma = 0$), the KH model ($\gamma = 0.1, 0.5$), and the CC model ($\gamma = 1$).

Fig 4 shows the simulation results of the Gini index $g$, total exchange $f$, and Kendall $\tau$ for saving rate $\lambda$ and surplus contribution rate $\gamma$ of the wealthy by using Eqs (5) to (7). Fig 4A to 4D show that, in general, as the saving rate $\lambda$ (stock) rises, the Gini index $g$ (disparities) and the

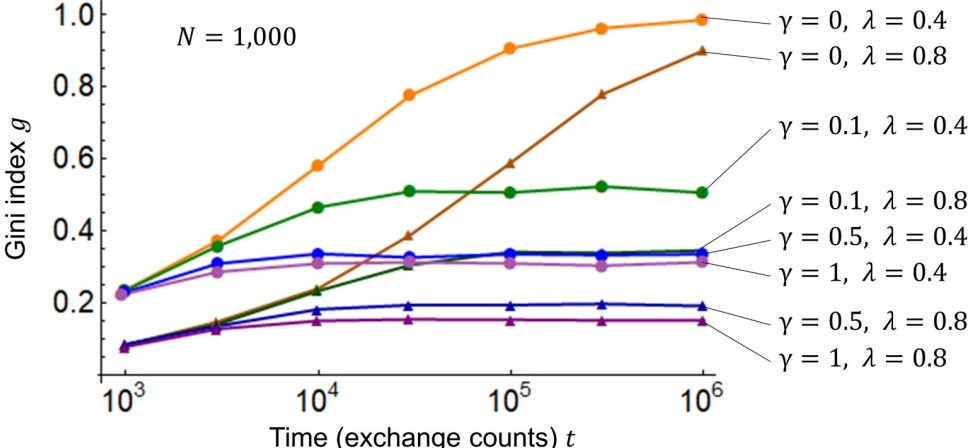

**Fig 3. Gini index $g$ for saving rate $\lambda$ and surplus contribution rate $\gamma$ of the wealthy.** The horizontal axis represents time $t$ while the vertical axis represents the Gini index $g$. The number of agents $N = 1,000$, and the initial values of assets $m_i(0) = 1$ are the same as those of Fig 2. Two different saving rates, $\lambda = 0.4, 0.8$, and four different surplus contribution rates of the wealthy, $\gamma = 0, 0.1, 0.5, 1$, for a total of eight cases, are simulated.

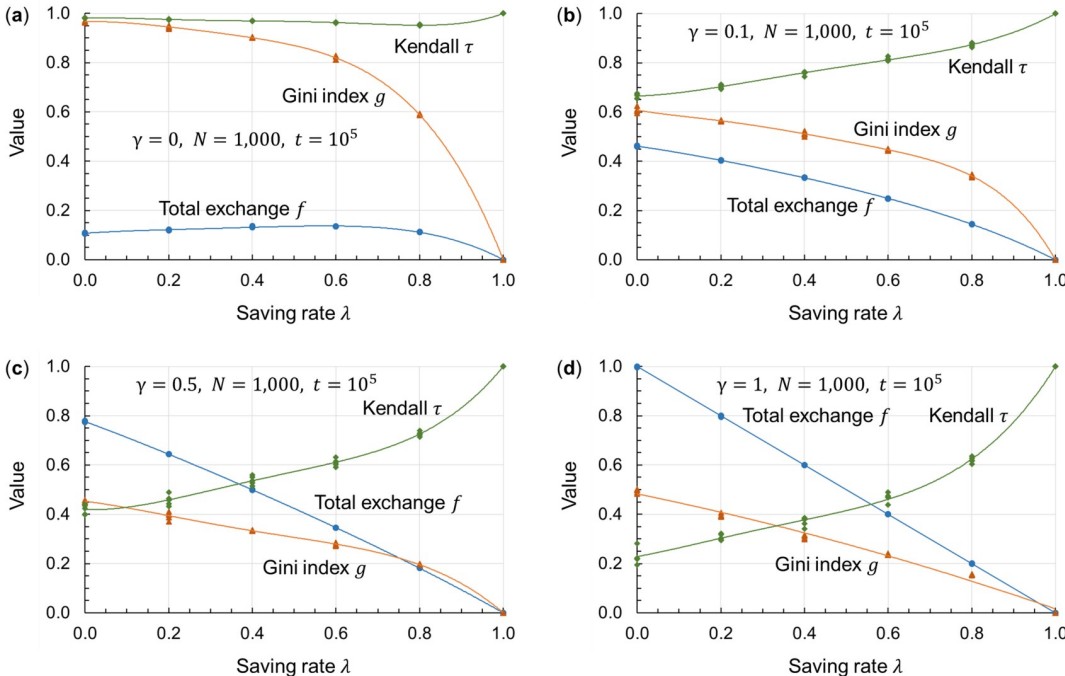

**Fig 4. Simulation results of $g$, $f$, and $\tau$ for $\lambda$ and $\gamma$.** The horizontal axis represents the saving rate $\lambda$. In (a) to (d), the number of agents is $N = 1,000$, and the times are $t_{max} = 10^5$, $t_1 = 9.9 \times 10^4$, and $t_2 = 10^5$. In the vicinity of $t = 10^5$, as seen in Fig 3, the Gini index and asset distribution generally converge even as the saving rate $\lambda$ differs except for the case wherein $\gamma = 0$. Under this assumption, the Gini index $g$, total exchange $f$, and Kendall rank correlation coefficient $\tau$ are simulated for the saving rate $\lambda$ with respect to four different surplus contribution rates of the wealthy: $\gamma = 0, 0.1, 0.5, 1$.

total exchange $f$ (flow) decline, while the rank correlation coefficient $\tau$ (disparities becoming permanent) increases. In other words, there is a trade-off between limiting disparities and vitalizing the market. A comparison between Figs 4A to 4D reveals that, as the surplus contribution rate $\gamma$ of the wealthy rises, the Gini index $g$ (disparities) and the rank correlation coefficient $\tau$ (disparities becoming permanent) tend to decline, and the total exchange $f$ (flow) tends to increase. Thus, the surplus contribution by the wealthy moderates the trade-off between limiting disparities and vitalizing the market.

Fig 5 is a three-dimensional graph that describes the changes in the Gini index $g$ and the total exchange $f$ when the saving rate $\lambda$ and the surplus contribution rate $\gamma$ of the wealthy are changed. When the saving rate $\lambda$ is changed, the Gini index $g$ and the total exchange $f$ change in a similar fashion. Thus, there is a trade-off between limiting disparities and vitalizing the market. When the surplus contribution rate $\gamma$ is changed, the Gini index $g$ and the total exchange $f$ change in the opposite direction. This means that disparities can be limited, even as the market is vitalized. Therefore, Fig 5 also shows that using the surplus stock of the wealthy will be more effective in limiting disparities than increasing savings.

In general, 0.4 is the warning level for the Gini index. Social unrest may occur if the value exceeds this level [3]. The KH model proposed herein is intended to examine the relative tendencies of wealth distribution and disparities. Thus, absolute figures are not very meaningful. Even so, it can still be argued that the surplus contribution rate $\gamma$ of the wealthy should be at least 0.4, roughly read from Fig 5, considering that the warning level for the Gini index is 0.4 and that the average saving rate relative to the global GDP is approximately 0.25 [26].

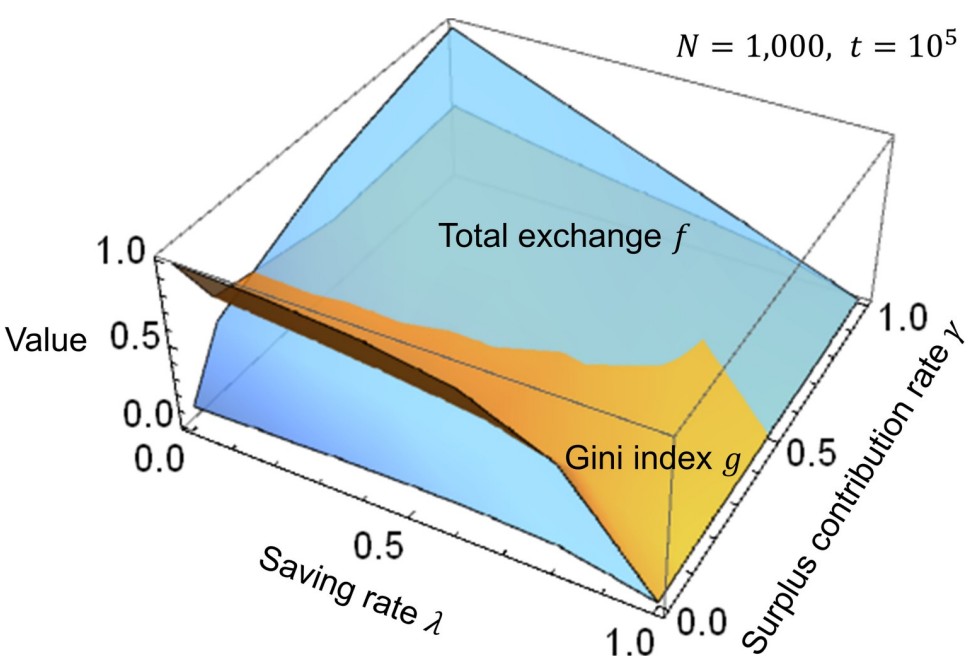

**Fig 5. Simulation results of _g_, _f_, and _τ_ for _λ_ and _γ_.** In the three-dimensional graph, the horizontal axes represent the saving rate _λ_ and the surplus contribution rate _γ_ of the wealthy, and the vertical axis represents the Gini index _g_ and the total exchange _f_. The number of agents is $N = 1,000$, and the times are $t_{max} = 10^5$, $t_1 = 9.9 \times 10^4$, and $t_2 = 10^5$, the same as in Fig 4.

## New equations between flow, disparities, saving, surplus stock, and metabolism

In the last part of this section, based on the calculations in Figs 4 and 5, a relational expression among four parameters—the saving rate _λ_, surplus contribution rate _γ_, total exchange _f_, and Gini index _g_—and a relational expression between two parameters—the total exchange _f_ and rank correlation coefficient _τ_—are derived. These equations were made possible by introducing parameters _f_ and _τ_ for the first time, which is an original contribution to the literature.

In Fig 6A, the horizontal axis is $(1-\lambda)\cdot\gamma$, and the vertical axis is $^f/_g$. In Fig 6B, the horizontal axis is _f_, and the vertical axis is _τ_. The former indicates an approximate equation among the four parameters—a new discovery shown in Eq (8). The latter shows an approximate equation between the two parameters—a new discovery shown in Eq (9). These two equations concisely express the relationship between and among the parameters explained in Figs 4 and 5.

$$\frac{f}{g} \sim \frac{1}{2} \cdot \ln(1 - \lambda) \cdot \gamma + 2, \tag{8}$$

$$\tau \sim -f. \tag{9}$$

Here, the meaning of Eq (8) is considered. When coefficients $^1/_2$ and 2 on the right side are put into the logarithm, the right side becomes $\ln e^2 \cdot \sqrt{(1 - \lambda) \cdot \gamma}$. In this case, $\sqrt{(1 - \lambda) \cdot \gamma}$ is a unitary amount of exchange in terms of dimension, while $e^2$ is a constant figure proportional to the overall asset amount (in this calculation, the initially established asset value of $m_i(0) \times N = 1 \times 1,000$). Thus, Eq (8) can be expressed as Eq (10)—which is a more general

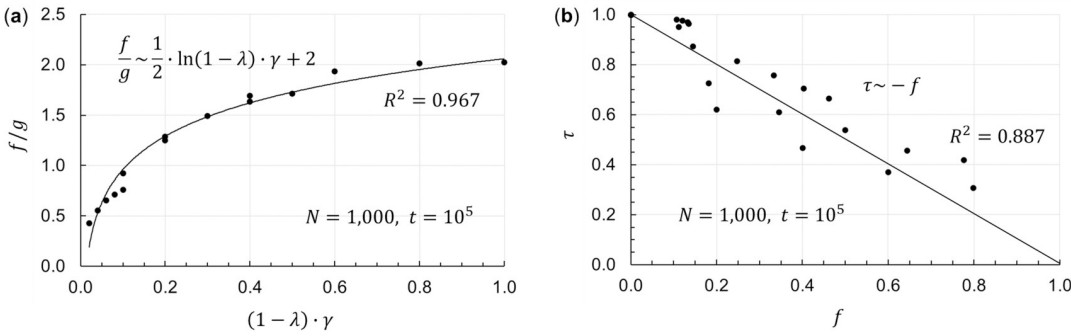

**Fig 6. Relationship between $\lambda$, $\gamma$, $f$, $g$, and $\tau$.** In (a), the horizontal axis is $(1-\lambda)\cdot\gamma$ of the saving rate $\lambda$ and the surplus contribution rate $\gamma$ of the wealthy, and the vertical axis is $f/g$ of the total exchange $f$ and the Gini index $g$. In (b), the horizontal axis is $f$, and the vertical axis is the Kendall rank correlation coefficient $\tau$. A substantial relationship can be seen for both (a) and (b) from the coefficient of determination $R^2$ of the regression equation. The number of agents is $N = 1{,}000$, and the times are $t_{max} = 10^5$, $t_1 = 9.9\times10^4$, and $t_2 = 10^5$, the same as in Figs 4 and 5.

equation, but newly derived—by setting the constant as $C$.

$$\frac{f}{g} \sim \ln\sqrt{(1-\lambda)\cdot\gamma} + C. \qquad (10)$$

The first term on the right side of Eq (10) is a natural logarithm. This may be because the KH model forms a gamma distribution; the probability density function of a gamma distribution is expressed by functions that include natural exponential functions, and this distribution is related to the total exchange $f$ and the Gini index $g$.

Note that if the society is compared to a living body [28], total exchange (flow) $f$ is vitality, rank correlation coefficient $\tau$ is metabolism, Gini index (disparities) $g$ is physical condition, saving rate $\lambda$ is fat, surplus contribution rate $\gamma$ is nutrition, and constant $C$ is physical strength originally possessed by the body. Eqs (9) and (10) show that if a body accumulates excess fat and has imbalanced nutrition, it loses energy, metabolism slows down, and its physical condition deteriorates. For the society—a kind of living body—to remain healthy and full of vigor, it is necessary to distribute nutrients (wealth) throughout the body.

Finally, we investigate whether the actual statistical data from various countries indicate a relationship identical to that in Eq (10). An analysis was conducted on the 37 member countries of the Organisation for Economic Co-operation and Development (OECD). The next section shows the procedure for finding the four parameters—total exchange $f$, Gini index $g$, saving rate $\lambda$, and surplus contribution rate $\gamma$—for each country.

We use World Bank Open Data [26, 29–31]. The GDP per capita (in current US dollars) [29] is regarded as the total exchange $f$, Gini index (the World Bank estimate) [30] as the Gini index $g$, gross savings (as a percentage of GDP) [26] as the saving rate $\lambda$, and tax revenue (as a percentage of GDP) [31] as the surplus contribution rate $\gamma$. While tax revenue is not a surplus contribution rate *per se*, it can be surmised that the use of surplus stock is progressing in a country where tax revenue is high. Table 1 shows the average value during the three-year period from 2017 to 2019 for each parameter, as well as the value of $(1-\lambda)\cdot\gamma$ and $f_{norm}/g$ calculated from these average values. $f_{norm}$ is the value obtained by normalizing $f$ with the maximum value $f_{norm}$ of Luxembourg.

Fig 7 is a graph that plots the value of $(1-\lambda)\cdot\gamma$ and $f_{norm}/g$ for each country, with the $x$ axis representing the former and the $y$ axis representing the latter (countries with missing data are excluded). There are variations in how these data are plotted. However, countries are classified

**Table 1. Data of Organisation for Economic Co-operation and Development member countries.**

| Country | GDP per capita (current US $) | Gini index (World Bank estimate) | Gross savings (% of GDP) | Tax revenue (% of GDP) | $(1-\lambda)\cdot\gamma$ | $\frac{f_{norm}}{g}\left(f_{norm}=\frac{f}{f_{max}}\right)$ |
|---|---|---|---|---|---|---|
| | $f$ | $g$ | $\lambda$ | $\gamma$ | | |
| Australia | 555 | – | 0.220 | 0.228 | 0.178 | – |
| Austria | 497 | 0.303 | 0.272 | 0.255 | 0.185 | 1.453 |
| Belgium | 461 | 0.273 | 0.251 | 0.234 | 0.175 | 1.493 |
| Canada | 459 | 0.333 | 0.198 | 0.129 | 0.103 | 1.219 |
| Chile | 153 | 0.444 | 0.188 | 0.178 | 0.144 | 0.304 |
| Colombia | 65 | 0.505 | 0.163 | 0.149 | 0.125 | 0.114 |
| Czech Republic | 225 | 0.250 | 0.268 | 0.149 | 0.109 | 0.799 |
| Denmark | 598 | 0.285 | 0.301 | 0.332 | 0.232 | 1.860 |
| Estonia | 225 | 0.304 | 0.286 | 0.210 | 0.150 | 0.655 |
| Finland | 484 | 0.274 | 0.236 | 0.208 | 0.159 | 1.566 |
| France | 403 | 0.320 | 0.231 | 0.242 | 0.186 | 1.115 |
| Germany | 463 | – | 0.285 | 0.115 | 0.082 | – |
| Greece | 196 | 0.337 | 0.092 | 0.259 | 0.235 | 0.516 |
| Hungary | 159 | 0.301 | 0.265 | 0.226 | 0.166 | 0.468 |
| Iceland | 704 | 0.261 | 0.234 | 0.233 | 0.179 | 2.387 |
| Ireland | 757 | 0.314 | 0.348 | 0.182 | 0.119 | 2.134 |
| Israel | 420 | 0.000 | 0.245 | 0.234 | 0.177 | |
| Italy | 334 | 0.359 | 0.209 | 0.245 | 0.194 | 0.824 |
| Japan | 393 | – | 0.280 | – | – | – |
| Korea, Rep. | 323 | – | 0.359 | 0.150 | 0.096 | – |
| Latvia | 171 | 0.354 | 0.229 | 0.226 | 0.174 | 0.429 |
| Lithuania | 186 | 0.365 | 0.204 | 0.178 | 0.141 | 0.450 |
| Luxembourg | 1130 | 0.350 | 0.172 | 0.260 | 0.215 | 2.861 |
| Mexico | 96 | 0.454 | 0.235 | 0.131 | 0.100 | 0.188 |
| Netherlands | 514 | 0.283 | 0.315 | 0.234 | 0.160 | 1.606 |
| New Zealand | 426 | – | 0.212 | 0.282 | 0.222 | – |
| Norway | 776 | 0.273 | 0.344 | 0.231 | 0.152 | 2.514 |
| Poland | 150 | 0.300 | 0.197 | 0.172 | 0.138 | 0.443 |
| Portugal | 228 | 0.337 | 0.184 | 0.224 | 0.183 | 0.599 |
| Slovak Republic | 187 | 0.250 | 0.218 | 0.186 | 0.146 | 0.663 |
| Slovenia | 252 | 0.244 | 0.267 | 0.184 | 0.135 | 0.914 |
| Spain | 294 | 0.347 | 0.225 | 0.139 | 0.108 | 0.749 |
| Sweden | 533 | 0.294 | 0.288 | 0.278 | 0.198 | 1.605 |
| Switzerland | 818 | 0.329 | 0.333 | 0.101 | 0.068 | 2.199 |
| Turkey | 97 | 0.417 | 0.266 | 0.172 | 0.126 | 0.206 |
| United Kingdom | 419 | 0.351 | 0.136 | 0.255 | 0.221 | 1.057 |
| United States | 628 | 0.413 | 0.187 | 0.106 | 0.086 | 1.345 |

SOURCE.—World Bank Open Data (https://data.worldbank.org/).

NOTE.—blue rows: high GDP per capita, green rows: middle GDP per capita.

into three groups based on their GDP per capita: high, middle, and low. As a result, a relationship similar to that in Eq (10) is observed within each group. These three groups are different from one another with respect to the constant *C* in Eq (10). The differences among countries within each group may reflect differences in social security policies. For each nation to expand

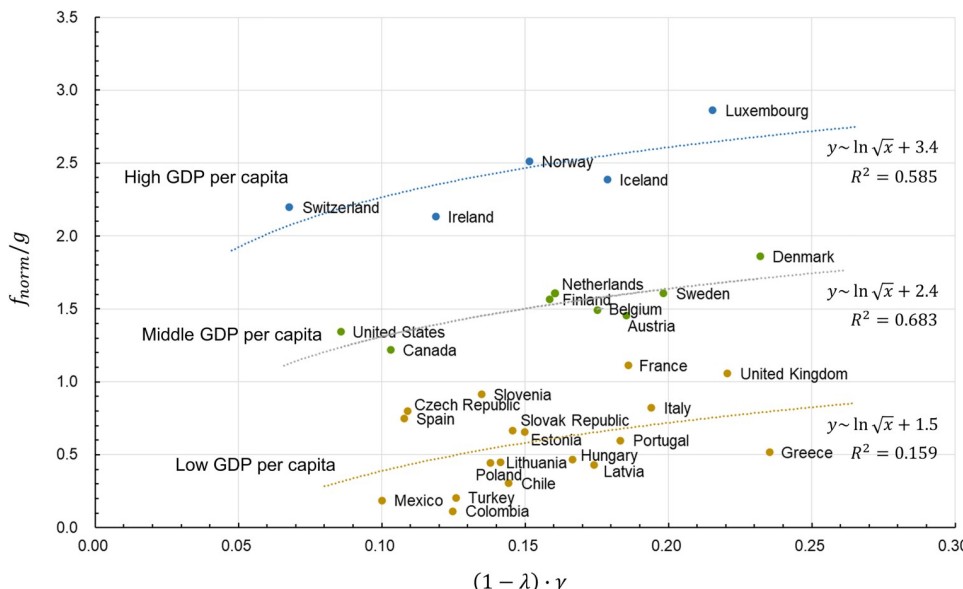

**Fig 7. Relationship between $\lambda$, $\gamma$, $f_{norm}$, and $g$ of OECD member countries.** The horizontal axis shows $(1-\lambda)\cdot\gamma$ of the saving rate $\lambda$ and the surplus contribution rate $\gamma$ of the wealthy, and the vertical axis is $f_{norm}/g$ of the total exchange $f_{norm}$ and the Gini index $g$, of countries from the Organisation for Economic Co-operation and Development (OECD). The blue, green, and brown plots show the groups of high, middle, and low GDP per capita, respectively. The high and middle GDP per capita groups fit well in the regression equations with high coefficients of determination $R^2$, but the low GDP per capita group has low $R^2$ because the regression equation was applied to all the remaining countries except the high and middle groups.

its $f_{norm}/g$, that is, activate flow and limit disparities, it must either develop new resources to increase $C$, the total asset amount, or use $\gamma$, the surplus stock of existing assets. Here, "assets" refers not only to economic assets but also to those from nature, environment, social capital, and culture. If they consider these factors, countries would not only be able to solve problems related to flow and disparities but also help improve people's well-being.

## Discussion

In comparison to the KH model, there is a model in which tax revenue is distributed when an exchange takes place [17] as well as a model in which insurance payouts are distributed to losers of an exchange [18]. These models, rather than using surplus stock, redistribute income (flow). These measures may reduce the Gini index (disparities) but do not increase the total exchange (flow). For this reason, the market does not become vitalized, the ranking remains the same, and metabolism does not occur easily (e.g., [32, 33]). In comparison to these tax and insurance models, the KH model uses the surplus stock of the wealthy.

What measures can be implemented to use the surplus stock of the wealthy? One measure is to release the idle surplus stock that does not contribute to flow. For example, idle stock can be redistributed to the poor through institutional intervention. The wealthy can also be required to provide their idle stock to the market, and the ownership of idle stock may be severed from the right to use the stock so that this right can be publicly owned even as the wealthy own the stock itself. Such proposals are based on the "social security through stock" [22, 23]. They may help stimulate market flow and metabolism while limiting disparities, as indicated by the simulation results discussed in the "Results" section.

There is yet another measure to promote the use of stock. This involves a common-ownership self-assessed tax (COST) [34]. Under this system, stock can be owned by those who assess their stock highly when filing a self-assessed tax return. They acknowledge that their stock has a high utility value. Under this system, the owner cannot prevent others from claiming the stock. For this reason, this measure may prevent stocks from remaining idle and promote flow and metabolism. The insights obtained from this study's KH model may validate the effectiveness of the COST system from the viewpoint of econophysics. In other words, the provision of surplus stock to the market through auction in the form of COST would involve increasing the surplus contribution rate $\gamma$ of the KH model. As suggested by Eq (9), this will vitalize the total exchange $f$, that is, economic flow.

There is also a measure, taken from a different vantage point, to promote investments in social businesses. This is the financial use of the surplus stock of the wealthy in the KH model. It guarantees Rawls' fair equality of opportunity [35] and Hiroi's equal opportunity [22, 23]. Such a measure will indirectly transfer stocks to the poor and the socially vulnerable, increase economic flow by social businesses, and prevent disparities from becoming permanent [36]. Thus, what is needed is corporate governance and preferential tax measures that will further promote companies' contribution to the United Nation's Sustainable Development Goals and financial institutions' environmental, social, and corporate governance investments and social impact investments [37, 38]. In the information industry, there is a movement toward digital democracy and "platform cooperativism" [39]. Platform corporatism is the idea that users who create value on a platform should own the platform and profit from it. According to the KH model, a monopoly of information assets leads to information disparities. Thus, it is important that information stock be made accessible and shared among people. Joint ownership of information assets, such as digital data and platforms, should also be considered [40].

As mentioned above, our insights, gained by observing the cause of disparities, may lend support to social measures such as redistribution, public ownership, and joint ownership of stock.

Going forward, as a specific measure to use stock, we will begin by working on a method to separate the ownership of idle stock from the right to use it because a sudden introduction of a measure to promote joint ownership or public ownership of stock may encounter strong resistance from owners. One possible measure is to strengthen taxation on the ownership of idle stock, reduce taxes if the owner grants the right to use the stock, and allow the owner to gain profits by selling the right to use the stock.

Specifically, we are considering taking up the issues of abandoned farmland in rural districts and the revitalization of urban areas. Indeed, abandoned farmland is becoming a global concern from the standpoint of carbon storage and biodiversity, in addition to problems associated with population decline and aging [41]. We may pursue studies on special laws designed to facilitate the use of abandoned farmland, subsidies to farmers so that farmland may be restored, the use of abandoned farmland to meet the needs of people who maintain homes in both urban and rural areas as the COVID-19 pandemic persists, and the renewal of farmland as insurance against a possible future food crisis.

Regarding the revitalization of urban areas, idle stock, such as vacant stores and empty buildings, will be targeted so that stakeholders can establish a universally shared goal and create new mutual interests [42]. During this process, based on the insights obtained from this study, it may be possible to conduct a more detailed examination through a policy simulation (e.g., [43]) and demonstrate that flow and metabolism will increase in urban areas if the right to use idle stock is granted to the public. The results of the examination may be useful as an assessment tool during a consensus-building process among stakeholders. The theoretical insights obtained from the KH model may be useful in establishing common goals toward

policy agreement and help reduce the cost and time required for assessment by defining the direction and scope of detailed examinations.

## Conclusions

We proposed a new model to gain insights for solving problems related to economic disparities, named the KH model, which has more flexibility in determining the exchange amount than prior asset exchange models (the KK and CC models) in the theoretical contribution of econophysics. Specifically, a new variable parameter, the surplus contribution rate $\gamma$, was introduced to the KH model, as an addition to the existing variable parameter, the saving rate $\lambda$. New evaluation parameters, namely, the total exchange $f$ and the rank correlation coefficient $\tau$, were also introduced to add to the existing evaluation parameter, the Gini index $g$. Note that the KH model provides mathematical modeling of the non-mathematical policy proposal made by Hiroi [22, 23].

A simulation using the KH model shows that to increase market flow ($f$) and metabolism (the opposite of $\tau$) while limiting disparities ($g$), it is important to use the surplus stock of the wealthy ($\gamma$) while limiting savings ($\lambda$). This can be briefly described by the approximation equations, $f/g \sim \ln \sqrt{(1 - \lambda) \cdot \gamma} + C$ and $\tau \sim -f$, which were first discovered from the simulation by introducing the original parameters $\gamma$, $f$, and $\tau$. These equations are extremely useful in that they explicitly indicate the fundamental relationships among flow, stock, savings, disparities, and metabolism.

However, for the sake of tractability, the KH model adopts the simplified assumptions that the overall asset amount is constant and that the assets are exchanged between two agents. Future research could consider asset production and collective exchange models. For example, asset production based on increasing returns may increase disparities to benefit the wealthy, while collective exchange based on multi-stakeholder consensus may reduce disparities to benefit the poor. The KH model also treats the surplus contribution rate of the wealthy with one parameter. It may be useful to divide the surplus contribution rate into multiple parameters; a study examining the illiquid asset economy distinguishes the marginal propensity among poor hand-to-mouth households, wealthy hand-to-mouth households, and non-hand-to-mouth households [44].

While the mathematical results and formulas regarding flow, stock, savings, disparities, and metabolism in this research have been yielded by numerical simulations using the KH model, we recommend a future analytical study. Similarly, while the theoretical insights obtained herein, as well as the formulae, may provide strong support for the redistribution and public sharing of stock, we do not provide any specific proposal on policy creation or effective policy implementation. Such an attempt is reserved for future empirical studies in economics or sociology.

## Acknowledgments

The authors received valuable advice from the Hitachi Kyoto University Laboratory of the Kyoto University Open Innovation Institute regarding how to pursue this research; the authors would like to express their deepest gratitude.

## Author Contributions

**Conceptualization:** Takeshi Kato, Yoshinori Hiroi.

**Formal analysis:** Takeshi Kato.

**Funding acquisition:** Takeshi Kato, Yoshinori Hiroi.

**Investigation:** Takeshi Kato.

**Methodology:** Takeshi Kato, Yoshinori Hiroi.

**Project administration:** Takeshi Kato.

**Software:** Takeshi Kato.

**Supervision:** Yoshinori Hiroi.

**Validation:** Takeshi Kato, Yoshinori Hiroi.

**Visualization:** Takeshi Kato.

**Writing – original draft:** Takeshi Kato.

**Writing – review & editing:** Takeshi Kato, Yoshinori Hiroi.

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
