## [Decision Letter · Decision Letter 0]

28 Sep 2021

PONE-D-21-26966Wealth disparities and economic flow: Assessment using an asset exchange model with the surplus stock of the wealthyPLOS ONE

Dear Dr. Kato,

Thank you for submitting your manuscript to PLOS ONE. After careful consideration, we feel that it has merit but does not fully meet PLOS ONE’s publication criteria as it currently stands. Therefore, we invite you to submit a revised version of the manuscript that addresses the points raised during the review process.

There are several points rised by the reviewers that should be consider by the authors before the manuscript is ready for publication. One point that want to mention is that equations are not introduced or commented properly along the paper. The reader expects also that variables and parameter are defined in the text when the reading of the material is clearer.

We look forward to receiving your revised manuscript.

Kind regards,

Rodrigo Huerta-Quintanilla, Ph. D

Academic Editor

PLOS ONE

Journal Requirements:

Reviewers' comments:

Reviewer's Responses to Questions

**Comments to the Author**

1. Is the manuscript technically sound, and do the data support the conclusions?

Reviewer #1: Yes

Reviewer #2: Yes

2. Has the statistical analysis been performed appropriately and rigorously? 

Reviewer #1: Yes

Reviewer #2: Yes

3. Have the authors made all data underlying the findings in their manuscript fully available?

Reviewer #1: Yes

Reviewer #2: Yes

4. Is the manuscript presented in an intelligible fashion and written in standard English?

Reviewer #1: Yes

Reviewer #2: Yes

5. Review Comments to the Author

Reviewer #1: The main aim of the research is actual and could extend significantly the literature and research in the field of wealth disparities. Still, there are necessary revisions as:

-the introduction section should be extended. It is too abstract. The authors should introduce the aim and objective of the research and make a presentation why is there a need for their research.

-the main idea of the article should be carefully explained, as it is hardly legible in its present form.

- the author should extend consider introducing a literature review section and survey more literature related to these aspects;

- an important benchmark for research is simply missing. Therefore, the literature list should be supplemented with selected items in this field and discussed in the literature review.

- the figures are poorly visible due to low resolution;

- consider combining the Results and Discussion section and discuss the results on an ongoing basis

- it is necessary to designate a section of conclusions clearly indicating the obtained results of the analysis.

Reviewer #2: wonder whether in the introduction the Authors could add some more references on the already knowledge on latest high-level papers.

Include an elaborate discussion explaining the exact significance of your findings.

Probably the quality of the pictures in figure 1 could be improved in the last version.

6. PLOS authors have the option to publish the peer review history of their article (what does this mean?). If published, this will include your full peer review and any attached files.

Reviewer #1: No

Reviewer #2: No

---

## [Author Response · Author response to Decision Letter 0]

8 Oct 2021

Dear Editor and Reviewers:

Thank you for your suggestions and comments on our manuscript titled “Wealth disparities and economic flow: assessment using an asset exchange model with the surplus stock of the wealthy.” We also appreciate the time and effort you and each of the reviewers have dedicated to providing insightful feedback on ways to strengthen our paper. We have incorporated changes that reflect the detailed suggestions you have so graciously provided. We hope that our edits and the responses satisfactorily address all the issues and concerns you and the reviewers have noted.

---

## [Decision Letter · Decision Letter 1]

18 Oct 2021

Wealth disparities and economic flow: Assessment using an asset exchange model with the surplus stock of the wealthy

PONE-D-21-26966R1

Dear Dr. Kato,

We’re pleased to inform you that your manuscript has been judged scientifically suitable for publication and will be formally accepted for publication once it meets all outstanding technical requirements.

Kind regards,

Rodrigo Huerta-Quintanilla, Ph. D

Academic Editor

PLOS ONE

Additional Editor Comments (optional):

Reviewers' comments:

Reviewer's Responses to Questions

**Comments to the Author**

1. If the authors have adequately addressed your comments raised in a previous round of review and you feel that this manuscript is now acceptable for publication, you may indicate that here to bypass the “Comments to the Author” section, enter your conflict of interest statement in the “Confidential to Editor” section, and submit your "Accept" recommendation.

Reviewer #1: All comments have been addressed

Reviewer #2: (No Response)

2. Is the manuscript technically sound, and do the data support the conclusions?

Reviewer #1: Yes

Reviewer #2: (No Response)

3. Has the statistical analysis been performed appropriately and rigorously? 

Reviewer #1: Yes

Reviewer #2: (No Response)

4. Have the authors made all data underlying the findings in their manuscript fully available?

Reviewer #1: Yes

Reviewer #2: (No Response)

5. Is the manuscript presented in an intelligible fashion and written in standard English?

Reviewer #1: Yes

Reviewer #2: (No Response)

6. Review Comments to the Author

Reviewer #1: The authors have address to all recommendations. The manuscript could be considered for publication.

Reviewer #2: (No Response)

7. PLOS authors have the option to publish the peer review history of their article (what does this mean?). If published, this will include your full peer review and any attached files.

Reviewer #1: No

Reviewer #2: No

---

## [Editor Report · Acceptance letter]

26 Oct 2021

PONE-D-21-26966R1 

Wealth disparities and economic flow: Assessment using an asset exchange model with the surplus stock of the wealthy 

Dear Dr. Kato:

I'm pleased to inform you that your manuscript has been deemed suitable for publication in PLOS ONE. Congratulations! Your manuscript is now with our production department. 

Kind regards, 

on behalf of

Dr. Rodrigo Huerta-Quintanilla 

Academic Editor

PLOS ONE